# A High Performance Polyacrylonitrile Composite Separator with Cellulose Acetate and Nano-Hydroxyapatite for Lithium-Ion Batteries

**DOI:** 10.3390/membranes12020124

**Published:** 2022-01-20

**Authors:** Weiping Chen, Xiang Wang, Jianyu Liang, Yao Chen, Wei Ma, Siyuan Zhang

**Affiliations:** 1School of Materials Science and Engineering, Wuhan University of Technology, Wuhan 430070, China; 290563@whut.edu.cn (W.C.); 290782@whut.edu.cn (Y.C.); 290883@whut.edu.cn (W.M.); 303961@whut.edu.cn (S.Z.); 2Department of Mechanical Engineering, Worcester Polytechnic Institute, 100 Institute Road, Worcester, MA 01609, USA

**Keywords:** separator, polyacrylonitrile, nano-hydroxyapatite, lithium-ion batteries

## Abstract

The traditional commercial polyolefin separators suffer from high-temperature thermal shrinkage, low electrolyte wettability and other issues. In order to improve the overall performance of the separators, electrostatic spinning technology was applied to obtain PAN nanofiber separators with an average diameter of 320 nm. Then cellulose acetate (CA) resin and nano-hydroxyapatite (HAP) were introduced to fabricate the PAN/CA/HAP composite separators through the constant temperature hot pressing and dip-coating crafts. The composite separator has a good thermal stability, with no significant dimensional change after a constant temperature treatment of 200 °C for 35 min. The electrolyte uptake rate of the PAN/CA/HAP-1.0 composite separator reaches 281%, which exhibits an efficient ionic conductivity. At the same time, it also attains a tensile strength of 11.18 MPa, which meets the requirement for separator use. Button cells assembled from PAN/CA/HAP-1.0 composite separators have an excellent rate of performance (160.42 mAh·g^−1^ at 0.2 C) and cycle capability (157.6 mAh·g^−1^ after 50 cycles at 0.5 C). The results support that lithium-ion batteries assembled with PAN/CA/HAP-1.0 composite separators will exhibit higher safety stability and better electrochemical performance than that of polyolefin separators, with a very immense potential for application.

## 1. Introduction

Rapid economic development has resulted in the consumption of a great deal of non-renewable energy resources such as oil and gas, posing a huge challenge to the ecological environment and economic sustainability [1,2,3]. It is imperative to develop and apply clean energy technologies. The key components of lithium-ion battery include anode material, cathode material, electrolytes, a separator and its external materials. It occupies a crucial position as one of the significant electric energy storage research fields, with high energy reserve efficiency and a steady cyclic charge/discharge performance. Nowadays, lithium-ion batteries have been used in a large number of new energy vehicles, mobile phones, portable wearable devices and other technical products [4,5,6,7,8,9]. However, lithium-ion batteries are prone to fire and explosion, so more safety and reliability demands of lithium-ion batteries are put forward [10,11]. The main function of the separator is to isolate the anode from the cathode, allowing lithium ions to be transported freely back and forth, which plays an important role in ensuring the safety of the lithium-ion battery [12]. Due to the absence of polar groups in the molecular structure of conventional commercial polyolefin separators such as polyethylene (PE) and polypropylene (PP) separators, which are highly hydrophobic, the electrolyte can’t sufficiently and effectively infiltrate them [13]. Therefore, the ionic conductivity of lithium-ion batteries assembled with commercial polyolefin separators is not ideal. In addition, polyolefin separators fail in dimensional stability at temperatures above 140 °C with serious shrinkage and curling, which will cause security risks [14]. That is why it is vital to develop separators with a better performance and higher security.

In recent years, many researchers have replaced polyolefin separators by using high performance polymers as the backbone, and introducing binders and nano-active particles to prepare composite separators [15,16,17,18,19,20,21]. High performance polymer separators, with powerful polarity and adequate electrolyte infiltration, can greatly enhance the ionic conductivity of lithium-ion batteries, thus improving electrochemical performance. They also have high temperature dimensional stability, broadening the safe temperature range of lithium-ion batteries. Common substrates include polymers such as polyethylene oxide (PEO) [22], vinylidene fluoride-hexafluoropropylene copolymer (PVDF-HFP) [23], polyacrylonitrile (PAN) [24] and polymethyl methacrylate (PMMA) [25]. Kim et al. [26] dissolved PAN powder in DMF to obtain a 10 wt% solution, and prepared PAN porous separators by the phase inversion method, after dipping the PAN separators in EC/DMC electrolytes with three different lithium salts to achieve GPE, the maximum ionic conductivity was measured to be 2.8 mS/cm. J Lee et al. [27] used electrostatic spinning technology and a thermal imidization process to produce the PI separator, whose diameter was 300 nm, and also brushed Al_2_O_3_ nanoparticles on both sides of it, which exhibited an excellent charge/discharge behavior. Chen et al. [19] used electrostatic spinning technology to produce the PAN separator, and utilized phenoxy resin as the binder and zeolite (ZSM-5) as the active particles to get the PAN composite separator, which exhibited exceptional mechanical strength and low dimensional shrinkage with a discharge capacity of 152.5 mAh·g^−1^.

Researchers have shown that the separator made by electrostatic spinning technology has good heat resistance and high porosity, which will absorb more electrolytes [28,29]. The molecular structure of PAN also contains the nitrile group (-CN) that can interact with the lithium salt of the electrolyte [30]. Hence, PAN is a well suited material for a lithium-ion battery separator. However, the mechanical strength of a PAN nano-fiber separator is too weak to meet the tensile strength requirement of 10 MPa [31]. Nowadays, researchers usually utilize advanced methods such as blending, multiple composite stacking [32], coaxial electrostatic spinning [33], impregnated polymers [34], and electrostatic spraying [35] to improve the mechanical performance of separators which are fabricated by electrostatic spinning. Cellulose acetate (CA) [36] is one of the thermoplastic polymers that not only acts as a binder, but also contains a large number of hydroxyl groups (-OH) in its molecular structure, improving the wettability of the separator with the electrolyte effectively. Lee HJ et al. [37] made the CA-CaO composite separator by using CA and calcium oxide (CaO), and then coated it on the surface of the PP separator, which made its thermal stability and electrochemical performance significantly enhanced. Nano-hydroxyapatite (HAP) [38] particles with a large specific surface area and high surface activity can increase the electrolyte absorption and promote the Li^+^ movement. 

In this study, to develop higher safety with excellent electrical property polymer-based separators for lithium-ion batteries, a pure PAN separator was made by electrostatic spinning technology under certain spinning conditions, and it was dipped in a certain concentration of CA solution for a period of time after high temperature hot pressing to improve the mechanical performance. Then, the PAN/CA/HAP composite separators were further fabricated by introducing different contents of HAP. CA resin can increase the mechanical performance of the separator, while HAP particles can enhance the electrochemical performance of lithium-ion batteries by improving the electrolyte absorption of separator. Compared with the PAN separator and commercial PP separator, the PAN/CA/HAP-1.0 composite separator shows better overall performance, which can compensate for the flaws of polyolefin separators to avoid fire accidents with a certain market application value.

## 2. Materials and Methods

### 2.1. Materials

Polyacrylonitrile (PAN, M_w_ = 150,000, purity ≥ 99.9%) was provided from Hefei Sipin Technology Co., Ltd. (Hefei, China). Acetone, *N*,*N*-Dimethylformamide (DMF) and *N*-Methylpyrrolidone (NMP) were purchased from Sinopharm Group Chemical Reagent Co., Ltd. (Beijing, China). Cellulose acetate (CA) was bought from Aladdin Chemistry Co., Ltd. (Shanghai, China). Nano-hydroxyapatite (HAP, particle size of 20 nm) was obtained from Nanjing Dulai Biotechnology Co., Ltd. (Nanjing, China). Electrolyte (1 mol/L LiPF_6_, EC/EMC/DMC (1:1:1, volume ratio)) was offered from Duoduo Chemical Technology Co., Ltd. (Suzhou, China). Lithium metal was supported by Tianjin Zhongneng Lithium Industry Co., Ltd. (Tianjin, China). Commercial PP separator (Celgard 2400, the thickness of 20 µm) was obtained from Shenzhen Yuancheng Electronics Co., Ltd. ( Shenzhen, China) which was chosen for comparison.

### 2.2. Preparation of Pure PAN Separator

First, the white PAN powder was placed in a vacuum drying oven at 60 °C for 12 h. After sufficient drying, an appropriate amount of PAN powder was dissolved in DMF, and the light yellow PAN spinning solution with 10 wt% was prepared after magnetic stirring at 60 °C for 8 h. A disposable syringe with a capacity of 10 mL was used to obtain an appropriate amount of the above spinning solution, and it was allowed to stand until the air bubbles in the spinning solution completely disappeared. PAN separator was prepared by electrostatic spinning machine at 16 kV with a speed of 0.09 mm/min and a receiving distance of 15 cm for 4 h. Subsequently, the PAN separator was cut into 10 cm × 10 cm samples, placed between stainless steel plates, tightened with clamps, and put into a blast drying oven at 120 °C for 35 min. Later the PAN separator samples were dried in a vacuum drying oven at 80 °C for 18 h to fully remove the residual DMF. The PAN separator is called PAN in subsequent descriptions.

### 2.3. Preparation of PAN/CA/HAP Composite Separator

Different weights of HAP particles and Polyethylene Glycol (dispersing agent, the degree of polymerization is 2000) (20:1, *w*/*w*) were taken and added to 200 mL of acetone, and with the ultrasonic cleaning machine dispersed for 2 h, HAP dispersions with 0 wt%, 0.5 wt%, 1.0 wt% and 1.5 wt%, respectively, were formed. The CA solutions with HAP particles were obtained by severally weighing 10 g of CA resin, adding them to each group of HAP dispersions mentioned above, and stirring magnetically at room temperature for 8 h. The PAN separators were dipped in each group of CA solutions for 30 min, then they were taken out and dried at room temperature until there was little liquid, and transferred to a vacuum drying oven at 80 °C for 12 h. The PAN/CA/HAP composite separators were designed as PAN/CA/HAP-0, PAN/CA/HAP-0.5, PAN/CA/HAP-1.0, PAN/CA/HAP-1.5, respectively. 

### 2.4. Characterization

Fourier transform infrared spectrum (FT-IR, Nexus670, Thermo Nicolet, Waltham, MA, USA) was used to characterize the molecular structures of PAN and PAN/CA/HAP with the range of 400–4000 cm^−1^. Scanning electron microscopy (SEM, JSM-IT300, JEOL, Tokyo, Japan) was applied to observe the surface morphologies of the separator samples, and they all needed to be dried and sprayed with a gold layer. The tensile strengths of all separator samples (50 mm × 20 mm) were tested by using an electronic universal material testing machine (Instron5967, Instron Pty Ltd., Norwood, MA, USA) with a slow speed of 5 mm·min^−1^. Contact angle measuring machine (OCA20, Dataphysics, Filderstadt, Germany) was employed to test the contact angles of all separator samples. The thermal performance of the separators was known by using a thermal gravimetric analyzer–differential scanning calorimeter (TGA–DSC, STA 449F3, Netzsch, Selb, Germany) with a test temperature range of 20–600 °C and a heating rate of 10 °C/min under the N_2_ protection. Moreover, the thermal shrinkage of separators was studied by putting them in a blast drying oven at different temperatures from 140 to 200 °C for 35 min.

The electrolyte uptake of the separators was achieved by immersing them in electrolyte for 2 h, and then calculated by the Equation (1):Electrolyte uptake (%) = (W − W_0_)/W_0_ × 100%(1)
where W_0_ was the original weight and W was the weight after being saturated with the electrolyte of separators.

The porosity of the separators was analyzed by soaking them in n-butanol solution for 1 h, and then calculated by the Equation (2):Porosity (%) = (M − M_0_)/(ρ *×* V)(2)
where M_0_ was the original weight and M was the weight after being infiltrated with the electrolyte of separators, ρ was the density of n-butanol solution, and V was the initial volume of separators.

A thickness gauge (CH-1-S, Shanghai liuleng, China) was applied to determine the separator’s thickness. Using a glove box, the CR2032 button cell was assembled in the order of stainless steel sheet/separator/stainless steel sheet, and then the ionic conductivity was gotten simply by utilizing an electrochemical workstation (Nova2, Metrohm-Autolab, Shanghai, China). Electrochemical impedance between separator and electrode was acquired by assembling button cell in the sequence of lithium metal/separator/lithium metal under the AC impedance spectroscopy procedure (EIS) of the electrochemical workstation. The test conditions were in the frequency range of 0.1 Hz to 1 MHz and the voltage variation rate of 5 mV·s^−1^. The ionic conductivity was calculated by the Equation (3): σ = h/(R × S)(3)
where h was separator’s thickness, R was the bulk resistance and S was the area of stainless steel sheet.

Electrochemical stability window was tested by assembling button cell in the sequence of lithium metal/separator/stainless steel sheet under the liner sweep voltammetry from 2.5 V to 6.0 V at 5 mV·s^−1^. Cycle and rate performances were researched by assembling button cell in the order of lithium metal/separator/LiFePO_4_ cathode with a battery-testing equipment (CT2001A, LAND Electronics, Wuhan, China). Cycle performance was conducted at a charge/discharge rate of 0.5 C/0.5 C for 50 cycles, and rate performance was examined at the charge/discharge rates of 0.2 C/0.2 C, 0.5 C/0.5 C, 1 C/1 C and 2 C/2 C with 2.5–4.2 V for 5 cycles.

Each cell consumes approximately 200 µL of electrolyte in a glove box filled with argon gas.

## 3. Results and Discussion

### 3.1. FT-IR Analysis

FT-IR spectrums of the pure PAN separator and PAN/CA/HAP composite separator were subjected to prove the smooth incorporation of CA resin and HAP particles as shown in Figure 1. In the infrared spectrum curve of PAN, the peaks at 2242 cm^−1^ and 1450 cm^−1^ can be given in the stretching vibration of the -CN bond and the bending vibration of -CH_2_- [39]. In the infrared spectrum curve of the PAN/CA/HAP composite separator, the peaks at 3477 cm^−1^, 2940 cm^−1^ and 1751 cm^−1^ are attributed to the characteristic absorption peak of -OH, the stretching vibration of -CH_2_- and the strong symmetric stretching vibration of C=O in -COCH_3_, respectively, which indicates that CA resin was successfully introduced into the PAN separator [37]. Meanwhile, the peaks at 565 cm^−^^1^, 603 cm^−^^1^ and 1047 cm^−^^1^ are vested in the characteristic absorption peak of ${PO}_{4}^{3-}$, the absorption peaks of ${CO}_{3}^{2-}$ appeared at 1370 cm^−1^ and 1457 cm^−1^, further indicating that the HAP particles were successfully introduced into it [40].

### 3.2. Micromorphological Analysis

Figure 2 presents the microscopic morphologies of the PAN separator, PAN/CA/HAP composite separators with different HAP contents, and the elemental surface sweep of the PAN/CA/HAP-1.0 composite separator. As can be seen from Figure 2a, pure PAN separator has a uniform fiber distribution with an average diameter of 320 nm, high porosity and no beads on its surface, which can effectively meet the reciprocal transportation of Li^+^. But the lap between fiber filaments is loose, resulting in weak mechanical strength. Figure 2b–e show the surface structures of separators with the further introducing of CA resin and HAP particles. The introduction of the appropriate amount of CA resin can increase the adhesiveness of fiber filaments to improve the mechanical performance of the PAN separator (Figure 2b). With the increase in HAP concentration, the amount of HAP particles on the surfaces of separators also gradually rises, and due to the small diameter of HAP particles and high surface activity, their Li^+^ transmission performance can be apparently improved to some extent (Figure 2c,d). However, the excess of HAP causes agglomeration of particles, which results in the blockage of micropores, the reduction of Li^+^ migration channels, and the obstruction of electrochemical reactions (Figure 2e). Figure 2f–i display the elemental distribution on the surface of the PAN/CA/HAP-1.0 composite separator. It can be seen that C, O, Ca and P elements are uniformly distributed among the fiber filaments, further indicating that HAP particles are successfully introduced into the separator [38].

### 3.3. Porosity and Electrolyte Uptake Analysis

The porosity and electrolyte uptake of different separators are shown in Figure 3. Commercial PP separator has small pores with high hydrophobicity, whose porosity and electrolyte uptake are the minimum, at 39% and 170%. With large pores and excellent electrolyte compatibility, the electrospun PAN separator has a porosity and electrolyte uptake of 70% and 281%. The porosity and electrolyte uptake of the PAN/CA/HAP-1.0 composite separator is 61% and 268%, respectively. Although the introduction of CA resin and appropriate amounts of HAP particles will partly fill the micropores, the surface structures of them are loose and porous with strong hydrophilicity. Moreover, the electrolyte can be adsorbed, not only in the micropores between the fiber filaments, but also in the CA resin and HAP particles, thus maintaining the good wettability of the electrolyte. However, when excessive HAP particles are introduced, the micropores are severely blocked and the electrolyte uptake drops. The porosity of the PAN/CA/HAP-1.5 composite separator is 51%, while the electrolyte uptake is only 208%, which will retard the normal movement of Li^+^.

### 3.4. Mechanical Performance Analysis

Figure 4 shows the stress–strain curves of different separators. The maximum tensile strength of the electrospun PAN separator is only 4.71 MPa, which fails to meet the minimum requirement of 10 MPa [31] for the winding machine. The addition of CA resin can clearly enhance the interaction between nanofibers, and effectively improve the tensile strength of the PAN separator to 13.42 MPa. With the increase of HAP content, the mechanical performance of separators gradually decreased, but the tensile strength of the PAN/CA/HAP-0.5 and PAN/CA/HAP-1.0 composite separators remained at 11.64 MPa and 11.18 MPa, respectively, which still met the practical use requirement of a lithium-ion battery separator. The main reason for the decrease in the mechanical performance of the composite separators was that during the dip-coating, part of the CA resin no longer adhered between the fiber filaments, but surrounded the surfaces of the HAP particles, thus weakening the tensile strength.

### 3.5. Ionic Conductivity Analysis

Figure 5 shows the electrochemical AC impedance spectra of CR2032 button cells assembled with different separators. The bulk resistance (R) is the intercept of the curve in the figure with the horizontal axis. The bulk resistances of the button cells, assembled with the PP, PAN and PAN/CA/HAP-0 separators are 1.275 Ω, 1.078 Ω and 1.010 Ω, respectively, and the ionic conductivities are calculated to be 0.86 mS·cm^−1^, 1.76 mS·cm^−1^ and 2.21 mS·cm^−1^. Apparently, the Li^+^ migration efficiency of button cells assembled by the commercial PP separator is relatively low because of the absence of polar groups in its molecular structure and the small and dilute micropores, leading to a slower electrolyte uptake. The best bulk resistance of 0.738 Ω and ionic conductivity of 3.02 mS·cm^−1^ are achieved for the button cells assembled by the PAN/CA/HAP-1.0 composite separator. The surface polarity and electrolyte wettability of the PAN separator are enhanced because of the introduction of CA resin and HAP particles, which makes the ionic conductivities of assembled button cells show a great improvement despite the decrease in porosity. Considering satisfactory microstructure, credible porosity, sufficient electrolyte wettability, suitable mechanical performance and excellent ionic conductivity, the PAN/CA/HAP-1.0 separator was chosen as a typical sample of the PAN/CA/HAP composite separators for subsequent studies. The ionic conductivities were calculated in Table 1.

### 3.6. Thermal Performance Analysis

The thermal performance of the separator is an important factor in measuring the safe operation of a lithium-ion battery. The thermal performances of the PP, PAN, PAN/CA/HAP-0 and PAN/CA/HAP-1.0 separators were researched using the DSC from Figure 6a. In the DSC curve, an obvious heat absorption peak is observed near 143 °C, which indicates that the PP separator melts at this temperature, while the PAN separator shows the first strong exothermic peak in the range from 270 °C to 330 °C, reaching a peak near 309 °C. Besides, both the PAN/CA/HAP-0 and PAN/CA/HAP-1.0 separators have the exothermic peaks around 312 °C. Compared with the PAN separator, they have reduced exothermic peak areas, but increased peak temperatures. This is because the PAN molecular structure unit contains nitrile group (-CN), which exhibits greater rigidity, and a cyanide cyclization reaction and dehydrogenation reaction occur after 270 °C, releasing more heat in advance. The TG test is applied to further analyze the thermal performances of different separators, as shown in Figure 6b. In the TG curve, the PP separator doesn’t show massive weight loss until 450 °C. Although its initial decomposition temperature is the highest, it starts to melt at 110 °C. It is not suitable for use as a separator under a high temperature environment, and the weight% is only 0.42% after the TGA test. Furthermore, the final weight% of the PAN/CA/HAP-0 separator is only 33.38%, which is a worse thermal performance than that of the PAN separator. However, the weight% of the PAN/CA/HAP-1.0 separator is 68.94%, which indicates that the introduction of HAP particles can boost the thermal stability of the separator to some extent.

At the same time, it is also critical for a lithium-ion battery separator to maintain good dimensional stability in a high-temperature environment. Figure 7 shows the dimensional changes of the PP, PAN, PAN/CA/HAP-0 and PAN/CA/HAP-1.0 separators after heat treatment at 25 °C, 140 °C, 160 °C, 180 °C and 200 °C for 35 min, respectively. It can be seen that the PP separator a large thermal shrinkage takes place, which directly reflects its inferior high-temperature dimensional stability. In addition, the PAN separator only slightly shrinks after heat treatment at 200 °C, while the PAN/CA/HAP-0 and PAN/CA/HAP-1.0 separators always maintain dimensional consistency before and after heat treatment, ensuring that the anode and cathode of a lithium-ion battery don’t directly contact each other under high-temperature conditions. Obviously, the PAN composite separators with CA resin and HAP particles are expected to improve the safety of a lithium-ion battery.

### 3.7. Contact Angle Analysis

The transient contact angles of different separators with electrolyte are shown in Figure 8 to further research the wettability of separators. The contact angle of the PP separator is the largest at 52.5°, indicating that it has the worst electrolyte compatibility, while the PAN separator has a contact angle of 37.1°, whose electrolyte wettability will be better than that of PP separator. Furthermore, with the introduction of CA resin and HAP particles, the contact angles of separators are reduced to 31.7° (PAN/CA/HAP-0) and 23.6° (PAN/CA/HAP-1.0), respectively, showing excellent wettability. This is because CA resin increases the hydrophilicity of separators. The porous surface of HAP particles, with a large specific surface area and the ability to interact with the Lewis acid-base of the electrolyte, also promotes the electrolyte uptake of them. Appendix A shows the contact angles of PAN/CA/HAP-1.0 separator with electrolyte at different times. With increasing time, the contact angle of PAN/CA/HAP-1.0 separator at different times of the electrolyte becomes progressively smaller. After 28.31 s, the contact angle was only 3.7°.

### 3.8. Electrochemical Performance Analysis

#### 3.8.1. Interfacial AC Impedance and LSV

Figure 9a shows the interfacial AC impedance spectra and equivalent circuit graph of button cells assembled with different separators. The first semicircle in the mid-high frequency region is the surface separator (SEI) impedance R_sf_, and another semicircle in the low frequency region is the charge transfer impedance R_ct_ at the electrode/electrolyte interface. Figure 9b shows the total resistance (R_sf_ + R_ct_) of the button cells, which is one of the significant factors affecting the charge/discharge performance. The total resistances of button cells assembled with the PP separator, PAN separator, PAN/CA/HAP-0 and PAN/CA/HAP-1.0 composite separators are 410.81 Ω, 302.06 Ω, 281.52 Ω and 233.49 Ω. It suggests that the PAN/CA/HAP-1.0 composite separator has the best interfacial compatibility with the electrode, which can absorb a large amount of electrolyte, slow down the damage of electrolyte to electrode materials, improve the interfacial stability and reduce the interfacial impedance. Figure 9c shows the linear sweep voltammetry curves (LSV) of button cells assembled with different separators. Their currents don’t change abruptly until 4.5 V; all button cells have the wide electrochemical stability windows, while the working voltage of common lithium battery is between 2.5–4.2 V, so they can all meet the steady use requirement. 

#### 3.8.2. Charge and Discharge Performance

Lithium-ion battery working principle: When charging, lithium-ions move through the electrolyte and separator to the negative electrode and are embedded in the internal structure of it. At this time, electrons can’t cross the separator, they can only pass through the external circuit, forming currents to the negative electrode and combining with Li^+^ to achieve electrical neutralization. When discharging, lithium-ions are released from the negative electrode (high potential) and embedded in the positive active material through the electrolyte and separator, and the electrons still pass through the external circuit to reach the positive electrode, realizing the conversion of chemical energy to electrical energy. 

The first cycle charge/discharge curves of full button cells assembled with different separators at 0.2 C are shown in Figure 10a for voltages in the range from 2.5 to 4.2 V. It can be seen that all batteries have the flat and wide charge/discharge platforms, which maintains a stable voltage. Meanwhile, the discharge capacities of button cells assembled with the PP separator, PAN/CA/HAP-0 and PAN/CA/HAP-1.0 separators are 144.9 mAh·g^−1^, 153 mAh·g^−1^ and 161 mAh·g^−1^ at the first cycle. The use of a composite separator can raise the charge/discharge performance of a lithium-ion battery. The rate performances of button cells assembled with different separators for 5 cycles at 0.2 C, 0.5 C, 1 C, 2 C and 0.2 C, respectively, are displayed in Figure 10b. The discharge capacities of all batteries decline with an increasing rate and tend to be consistent, which is the result of the polarization of the electrodes due to an increasing current density when the rate is increased during the charge/discharge process. At 0.2 C, the average discharge capacity of the button cells assembled with the PAN/CA/HAP-1.0 composite separator is the largest at 160.42 mAh·g^−1^, which indicates that the discharge performance of a battery can be maximized by using the PAN/CA/HAP-1.0 composite separator under the same situation. Figure 10c shows the cycle performances of button cells assembled with different separators for 50 cycles at 0.5 C. The highest and lowest discharge capacities of the battery assembled with the PAN/CA/HAP-1.0 composite separator are 158.6 mAh·g^−1^ and 156.6 mAh·g^−1^, with a cycle deviation of only 1.28%, which exhibits a high cycle capacity and stability with a practical application value compared with others. Figure 10d shows the first cycle Coulombic efficiency and average Coulombic efficiency of the cycle performance test. It can be found that the battery assembled with the PAN/CA/HAP-1.0 separator exhibits the best Coulomb efficiency values with 95.26% of the first cycle Coulombic efficiency and 99.48% of the average Coulombic efficiency, which is an improvement of 2.03% and 4.12%, respectively, compared to the cells assembled by the PAN separator. Initially, as the SEI membrane is not fully formed on the surface of the negative electrode, the charge and discharge process of the battery is not stable. These results have provided further evidence that the introduction of CA resin and HAP particles can improve the charge and discharge performance and enhance the stability of the battery. 

## 4. Conclusions

In this study, the PAN separator was prepared by electrostatic spinning technology, and then CA resin and HAP particles were introduced by the dip-coating method to fabricate the PAN/CA/HAP-1.0 composite separator with a tensile strength of 11.18 MPa, which obviously improves the mechanical strength of the separator, compared to the pure PAN separator. The composite separator has excellent heat resistance and high temperature dimensional stability, extending the operating temperature range of the lithium-ion battery. Due to the strong hydrophilicity of CA resin and the large specific surface area and high surface activity of HAP particles, the electrolyte wettability of the PAN/CA/HAP-1.0 composite separator becomes higher, which makes the battery assembled by the PAN/CA/HAP-1.0 composite separator exhibit a higher ionic conductivity and lower interfacial impedance. During the charge/discharge performance test, the PAN/CA/HAP-1.0 composite separator assembled button cell also reaches a high discharge capacity of 161 mAh·g^−1^, while presenting an excellent cycle stability and a wide electrochemical stability window. It also significantly improves the Coulombic efficiency values of the battery, making the charge and discharge process more stable. In conclusion, the innovative composite separator has an excellent overall performance and a promising application in the lithium-ion battery.

## Figures and Tables

**Figure 1 membranes-12-00124-f001:**
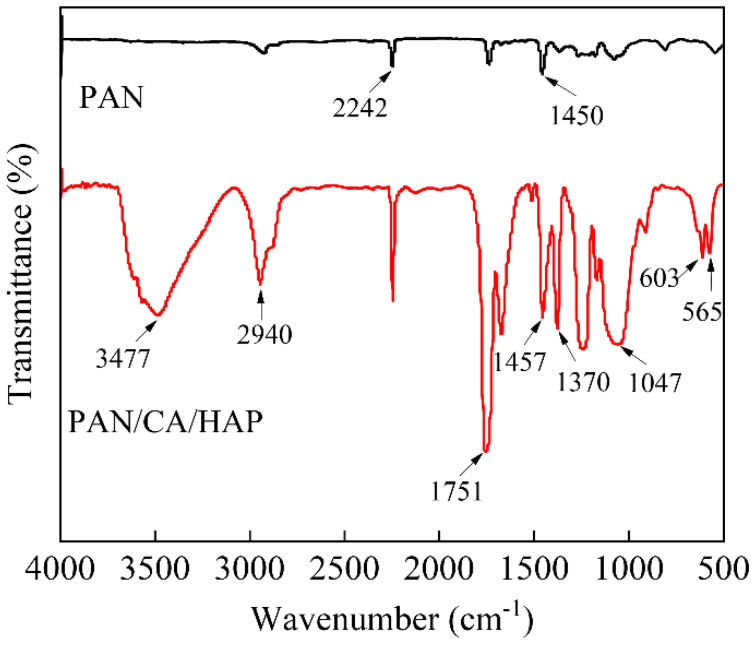
FT–IR spectra of pure PAN separator and PAN/CA/HAP composite separator.

**Figure 2 membranes-12-00124-f002:**
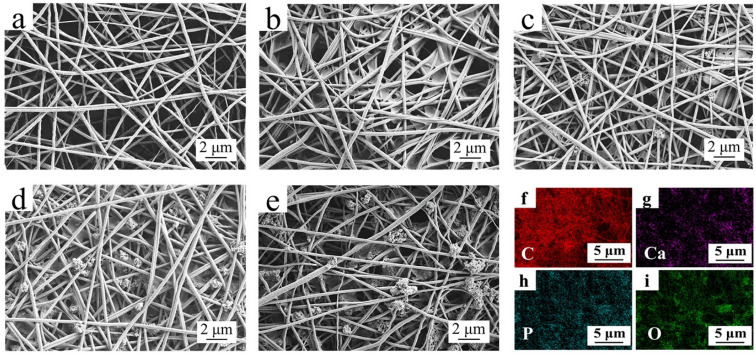
SEM micrographs for the surface of PAN separator and composite separators with different HAP particles concentrations: (**a**) PAN, (**b**) PAN/CA/HAP-0, (**c**) PAN/CA/HAP-0.5, (**d**) PAN/CA/HAP-1.0, (**e**) PAN/CA/HAP-1.5. (**f**–**i**) Elemental mapping of PAN/CA/HAP-1.0 composite separator.

**Figure 3 membranes-12-00124-f003:**
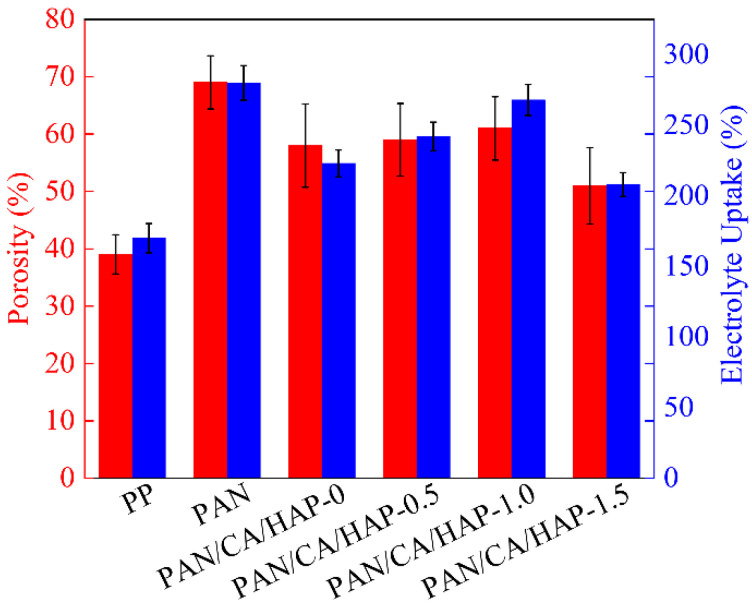
The porosity and electrolyte uptake of different separators.

**Figure 4 membranes-12-00124-f004:**
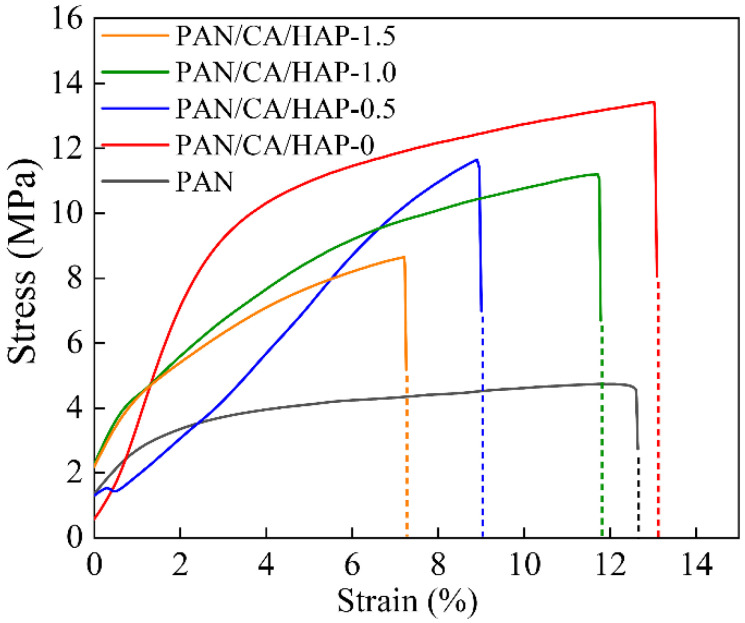
The stress–strain curves of different separators.

**Figure 5 membranes-12-00124-f005:**
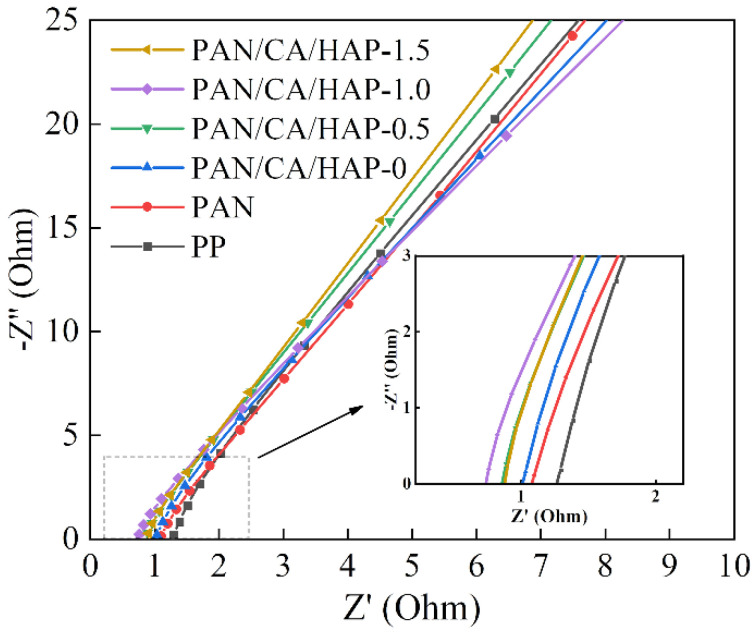
The electrochemical AC impedance spectra of CR2032 button cells assembled with different separators.

**Figure 6 membranes-12-00124-f006:**
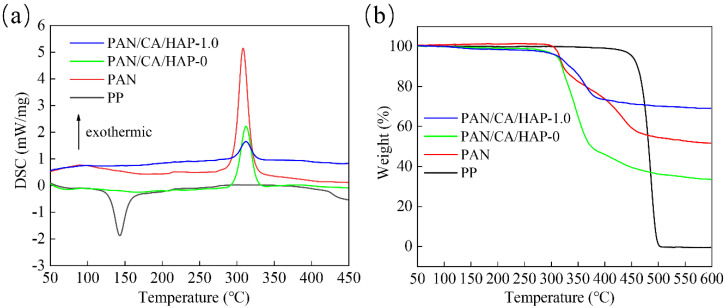
(**a**) DSC and (**b**) TG curves of PP, PAN, PAN/CA/HAP-0 and PAN/CA/HAP-1.0 separators.

**Figure 7 membranes-12-00124-f007:**
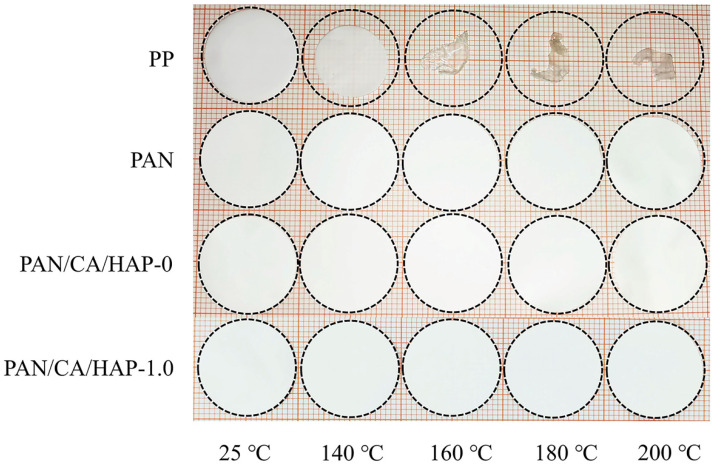
Thermal shrinkage behaviors of the separators after heat treatment at different temperatures for 35 min.

**Figure 8 membranes-12-00124-f008:**
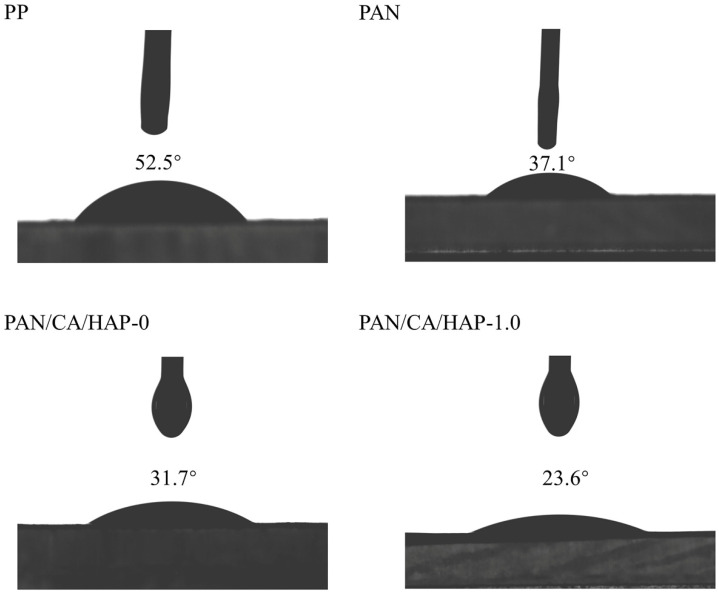
Contact angle test of PP, PAN, PAN/CA/HAP-0 and PAN/CA/HAP-1.0 separators with electrolyte.

**Figure 9 membranes-12-00124-f009:**
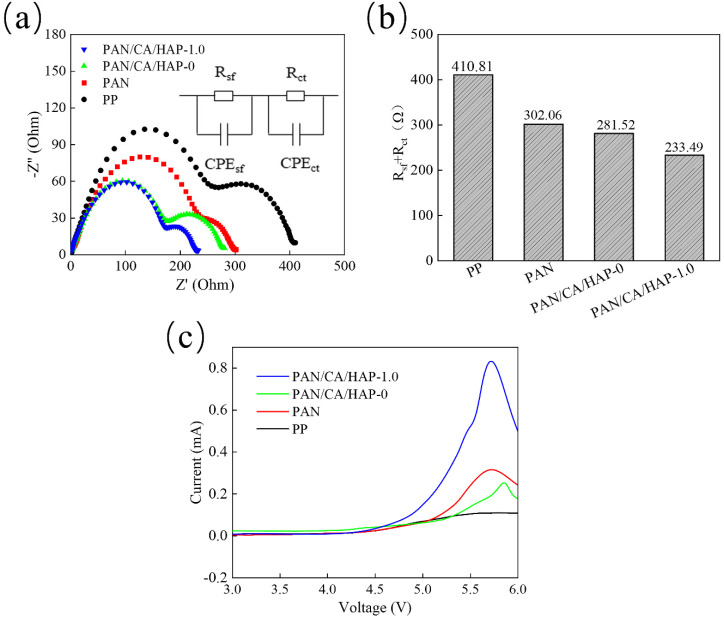
(**a**) Interfacial AC impedance spectra and equivalent circuit; (**b**) The sum of R_sf_ + R_ct_; (**c**) Linear scan voltammo–grams (LSV) curve.

**Figure 10 membranes-12-00124-f010:**
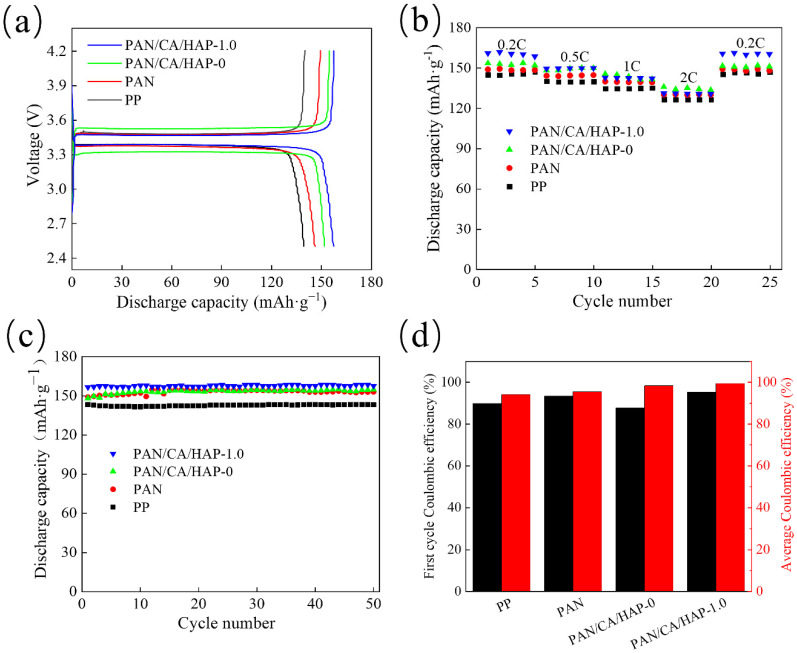
(**a**) Initial charge/discharge (0.2 C/0.2 C); (**b**) Rate performance (0.2 to 2 C); (**c**) Cycle performance (0.5 C); (**d**) Coulombic efficiency values of cycle test.

**Table 1 membranes-12-00124-t001:** The thickness and ionic conductivity of different separators.

Sample	Thickness, μm	Ionic Conductivity, mS·cm^−1^
PP	20	0.86
PAN	39	1.76
PAN/CA/HAP-0	46	2.21
PAN/CA/HAP-0.5	47	2.81
PAN/CA/HAP-1.0	46	3.02
PAN/CA/HAP-1.5	48	2.73

## Data Availability

The data presented in this study are available on request from the corresponding author.

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
