# Peer review of "A High Performance Polyacrylonitrile Composite Separator with Cellulose Acetate and Nano-Hydroxyapatite for Lithium-Ion Batteries"

_membranes, 2022, doi:10.3390/membranes12020124_

Round 1

Reviewer 1 Report

In this work the authors developed a series of polymer separators for lithium battery applications. Results are interesting and the studies performed in order to analyse these materials are convincing. The English language is of a sufficient level for publication. The selected topic fits with the journal request and it is in line with its broader readership. In summary, I recommend the publication of this manuscript in the Membranes journal only after a Minor Revision.

Here below are some comments:

  1. Some typos are present (e.g., missing spaces, a space between the temperature value and the °C unit, kV and not KV, etc.).
  2. Line 121: what is the exact role of PVA and is it removed? This point is not clear.
  3. Lines 150-154: are the separators swelling in n-butanol? If yes, this could compromise the porosity evaluation.
  4. Equation (3): typically, the ionic conductivity is indicated as “σ”, not “δ”.
  5. FT-IR studies: curves must be baseline-subtracted and normalized, in order to make easier the comparison.
  6. Lines 283 and 285: what is the carbon rate?
  7. DSC studies: is the exothermic event in PAN-based separator related to a reversible thermal transition, or is it due to the degradation of the material? This is because at the same temperature a weight loss is observed in TG experiments. In addition, could it be that the reduced area in PAN/CA/HAP-0 and PAN/CA/HAP-1.0 separators is due to the lower PAN content (i.e., part of the weight is occupied by CA and HAP).
  8. In the introduction, the authors should state also that the holy grail in this topic is to develop solid state electrolytes, polymeric or ceramic, in order to avoid fire and explosions (e.g., 10.1002/cssc.201500284, 10.1039/c8ee01053f, 10.1016/j.jpowsour.2018.07.118, 10.1039/c4cs00020j, 10.1016/j.electacta.2019.03.167, 10.1143/jjap.40.3246, 10.1038/s41428-020-00397-4, 10.1080/10587250210443). The use of highly stable separators is an interesting option, probably closer to the market, but not the final goal.
  9. Figure 9d: it is interesting to observe that the battery assembled using PAN/CA/HAP-0 has the largest overpotential, but still it is among the most performing. Could the authors discuss about this?
  10. Battery tests: did the authors tested the proposed materials more than one time? Did they confirm these results?

Reviewer 2 Report

Summary:

Lithium-ion batteries and related cell components have drawn significant attention in the last few years due to the fast development in the market and in science. One of the key components in the lithium-ion batteries is the separator that is used to block the contact of the positive and negative electrode, while keeping a smooth penetration of the liquid electrolytes. In this manuscript, the author conducted a systematic investigation using a polyacrylonitrile composite separator with cellulose acetate and nano-hydroxyapatite. The designed composite separator shows improved thermal stability and good electrolyte hosting. Generally, this is an interesting work and should be useful to the community. I would like to recommend its publication after addressing the comments and suggestions below.

  1. In the Materials and Methods, a thickness gauge is used for determining the separator thickness. However, it gives pressure to the soft polymer substrate. If it is possible, it is suggested to measure the thickness by the cross-sectional SEM observation.

  1. In the data presentation, the letters in most figures in the panels are too small to read. Again, most plots in the figures stack together, which is very hard to tell one from another.

  1. For the cycling performance analysis, it shows 50 cycles at C/2 rate. What would happen after 50 cycles?

  1. In both the rate performance and cycling performance, the discharge and charge reactions should be discussed. What are the Coulombic efficiency values of the cells?

  1. What are the amounts of electrolyte used in the cell? The information should be reported.

Round 2

Reviewer 2 Report

Authors have responded to the doubts and criticisms posed and done all corrections based on the reviewer comments.